# Single-Channel Blind Image Separation Based on Transformer-Guided GAN

**DOI:** 10.3390/s23104638

**Published:** 2023-05-10

**Authors:** Yaya Su, Dongli Jia, Yankun Shen, Lin Wang

**Affiliations:** School of Information and Electrical Engineering, Hebei University of Engineering, Handan 056038, China; yara_su@163.com (Y.S.);

**Keywords:** blind image separation, generative adversarial network, Transformer, UNet

## Abstract

Blind source separation (BSS) has been a great challenge in the field of signal processing due to the unknown distribution of the source signal and the mixing matrix. Traditional methods based on statistics and information theory use prior information such as source distribution independence, non-Gaussianity, sparsity, etc. to solve this problem. Generative adversarial networks (GANs) learn source distributions through games without being constrained by statistical properties. However, the current blind image separation methods based on GANs ignores the reconstruction of the structure and details of the separated image, resulting in residual interference source information in the generated results. This paper proposes a Transformer-guided GAN guided by an attention mechanism. Through the adversarial training of the generator and the discriminator, U-shaped Network (UNet) is used to fuse the convolutional layer features to reconstruct the structure of the separated image, and Transformer is used to calculate the position attention and guide the detailed information. We validate our method with quantitative experiments, showing that it outperforms previous blind image separation algorithms in terms of PSNR and SSIM.

## 1. Introduction

Image interference or pollution is very common in practical situations. In the same way that ears can distinguish the speaker from a noisy soundscape at a cocktail party [1], eyes can similarly distinguish between different objects in a complex visual situation. Blind image separation (BIS) obtains the source image according to the mixing image without knowledge of source distribution and mixing channel, which is what BSS does in computer vision. Single-channel blind image separation (SCBIS) is an extremely underdetermined BSS problem because the observed images are only available from a single receiver channel. The difficulty with the SCBIS problem lies in the high-quality recovery of multiple sources from the observed single image without knowledge of the transmission channel characteristics.

Independent component analysis (ICA) [2] assumes that source signals are independent of one another and non-Gaussian distributed. In robust principal component analysis (RPCA) [3], the dimensionality of the observed signals is reduced to separate the source signals. Non-negative matrix decomposition (NMF) [4] trains iteratively to decompose the mixture signal into two non-negative matrices: the weight matrix and the feature matrix. To acquire additional constraint information, the above methods make strong assumptions about the non-Gaussianity, independence, and non-negativity of the distribution of the source signal, which are not guaranteed in the image samples and result in a low-quality separated image.

GANs [5] are based on the idea of a zero-sum game, in which the generator and discriminator are treated as two faces of the game, and the generator and discriminator are continually optimized during the training process until the generated samples are able to successfully fool the discriminator. Combining the adversarial generative structures with other networks leads to a significant improvement in the quality of the images generated. By mapping information from random distributions to target domains via adversarial learning, GANs offer a solution to the problem of less priori in BSS.

Ref. [6] applied neurons to GANs and achieved one-to-many single-channel blind source separation by adjusting the number of neurons to map random signals to all domains of the source signal at the same time, but neuronal networks caused dimensional disasters when processing image data while losing spatial information. Ref. [7] trained the generators for each source signal, which mapped the randomness distribution onto the source signal domain, and then used maximum likelihood estimation to minimize the sum distance between the generated signals and the given mixture signal in order to separate the source signal. The two-step separation scheme caused a deviation between the source and generated signals. Ref. [8] used CYCLEGAN for separating and reconstructing two random mixture signals. This is equivalent to leaving one field of the generated separated signal unchanged and the other field swapped twice, such that the reconstructed mixed signal is identically distributed relative to the original mixture signal. The deep convolution networks resulted in the loss of valuable detail information. The above GANs focused only on mapping the mix to the source, ignoring the loss of detail caused by positional correlation, resulting in differences in texture representation between the separated image and the source.

Ref. [9] proposed the UNet structure, which links the convolution neural network into a U-shaped structure to gain context information, thereby reducing the valid feature loss. The Transformer framework proposed in [10] computes the association between different feature positions to obtain attentional information. Therefore, we use the generative adversarial network combined with the Transformer structure and UNet to guide the reconstruction of the separated image on the overall structure and detailed information.

### 1.1. Related Work

#### 1.1.1. BIS with UNet-GANs

UNet [9], as an applicable generative model, does a good job of reconstructing the target image by embedding the input image feature. Some SCBIS work has used GAN structures combined with UNet models. Ref. [11] treated the source and observed signals as information in different areas and solved the multisource single-channel blind split problem by combining multiple UNet-model-based DualGAN networks to train PDualGAN, but the detail representation of the source signal was not taken into account in the generation process. Ref. [12] combined the Attention module and UNet in the AGAN to achieve full-detail image generation; however, the attention model only focuses on the importance of single-pixel value while ignoring the interrelationship of pixels at different locations, leading to excessive attention on interference information in the input mix image. Ref. [13] cascaded two layers of GANs based on UNet, UGAN, and PAGAN to implement SCBIS in the case of a few samples, using UGAN for the generation of mixture samples and PAGAN for the separation of images. The resulting cascade GAN focuses more on the insufficient training samples than on the detailed information between pixels at different positions of the image.

#### 1.1.2. Transformer

Transformer [10] was originally proposed to compute the correlation between the input word vectors in machine translation. Refs. [14,15] introduced to the field of view by improving Transformer. Ref. [16] proposed a simple and extensible ViT model that splits images into a fixed number of patches and presented a multi-head self-attention mechanism for extracting feature information containing relational information. Transformer has shown excellent performance in various image detail processing. Refs. [17,18] used Transformer in an image restoration architecture to improve detail restoration by capturing pixel interaction relationships over long distances; Ref. [19] proposed UTNet, integrating Transformer architecture into convolutional neural networks to enhance finer detail segmentation through local information; Ref. [20] proposed CNN-Transformer for facial super-resolution tasks to improve fine facial details and enhance the fidelity and naturalness of reconstructed facial images.

Inspired by the detail-processing ability of Transformer and the extraction of global context features of UNetGAN, this paper proposes a Transformer-guided GAN to resolve the loss of valid information and to balance between the part and the whole. The GAN structure incorporates UNet for source reconstruction and a Transformer module for positional encoding and attention information extraction, improving separation accuracy and retaining detail representation.

## 2. Method

The overall structure of the Transformer-guided GAN is shown in Figure 1. The generator consists of a combination of a U-shaped connected convolution network with a Transformer structure. The discriminator is formed by the convolution neural network with the activation function only. For a given mix image $x\in R^{W\times H\times C},$ where W, H, C are its width, height, and number of channels, respectively, the goal of the generator is to reconstruct a high-quality separated image $\hat{s}\in R^{W\times H\times C}$ that is as similar as possible to the source image $s\in R^{W\times H\times C}$. The objective of the discriminator is to distinguish between s and $\hat{s}$. Through continuous adversarial learning of the generator and discriminator, the generator eventually reconstructs a plausible $\hat{s}$ for any given x.

### 2.1. Structure of the Generator

The structure is shown in detail in Figure 2. UNet consists of an M-layer convolution neural network connected by a U-shaped structure. Each layer uses a convolution kernel of size k and takes a step size of t for downsampling feature extraction of the input image, as shown in Equation (1):(1)$$f_{m}^{Conv}=Conv_{m}\left(x\right),$$
where $f_{m}^{Conv}\in R^{W_{m}{\times H}_{m}{\times C}_{m}}$ denotes the result of *m*-th feature extraction; $W_{m}=W//2^{t};\mathrm{and}\;\;H_{m}=H//2^{t}$. With the same setup as the convolution module, the deconvolution upsamples the m-th layer fusion feature $f_{m}$ back to the previous layer, as in Equation (2),
(2)$$f_{m-1}^{Deconv}=Deconv\left(f_{m}\right),$$
where $f_{m}\in R^{W_{m}{\times H}_{m}\times 2C_{m}}$ is the *m*-th layer fusion-feature concatenated convolution features and deconvolution features, as shown in Equation (3).
(3)$$f_{m}=Concat\left(f_{m}^{Conv},f_{m}^{Deconv}\right),$$
and $f_{m-1}^{Deconv}$ is the result of extracting the *m*-th fusion feature; when *m* = 1, $f_{m-1}^{Deconv}=f_{0}^{Deconv}$ denotes the generated separated image $\hat{s}$.

#### Transformer Scheme

Transformer connects the two ends of UNet. The structure we used in this paper, as shown in Figure 2, has a linear projection layer and an L-Layer Encoder.

As shown in Figure 3, after linear mapping, the convolution maps, the output of the UNet *M*-th convolution $f_{M}^{Conv}\in R^{W_{m}{\times H}_{m}\times 2C_{m}}$, is projected into N D-dimension patches, and then the position of each patch is encoded, as shown in Equation (4),
(4)$$P_{0}=\left[f_{0}E;\;f_{1}E;\cdots ;\;f_{N}E;\right]+E_{pos},$$
where $f_{n}\in R^{N\times (W_{m}\cdot H_{m})}$ denotes the 2D patches of $f_{M}^{Conv}$ and $N=C_{M}$; $E\;\in {\;R}^{(W_{m}\cdot H_{m})\times D}$ denotes the patch projection and $D=W_{m}\cdot H_{m}$; and $E_{pos}\in R^{N\times D}$ is the position embedding of each patch. 

The Encoder module first performs layer normalization on the patch features and then obtains the attention information through the multi-head attention mechanism as shown in Equation (5),
(5)$$P_{l}=MHA\left(LN\left(P_{l-1}\right)\right)+P_{l-1},$$
where *LN* is a de-correlation operation based on the mean and variance calculated in the channel dimension from the patch features of the mix image in the input batch, as shown in Equation (5),
(6)$$P_{LN}=\frac{P-E\left(P\right)}{Var\left(P\right)+\epsilon }\;*\;\gamma +\beta ,$$
where γ,β are trainable parameters. 

MHA is the key part for Transformer to explore the attention information, as shown in Figure 4. It first obtains the *Q*, *K*, *V* matrixes of the patch vectors by linear mapping as shown in Equation (7),
(7)$$Q,\;K,\;V=PW_{Q},\;PW_{K},\;PW_{V},$$
where $W_{Q},\;W_{K},\;\;\mathrm{and}\;W_{V}\in R^{N\times D}$ are trainable weight parameters. Then each matrix is divided and recombines them into h head vectors as shown in Equation (8),
(8)$$q_{i},\;k_{i},\;v_{i}=QW_{q}^{i},KW_{k}^{i},VW_{v}^{i},$$
where $W_{q}^{i}\in R^{N\times D_{h}},W_{k}^{i}\in R^{N\times D_{h}},\;W_{v}^{i}\in R^{N\times D_{h}},\;\mathrm{and}\;D_{h}=D//h$ are trainable weight parameters. The self-attention operation within each head vector calculates the correlation by the dot product of *q* and transposes *k* to get the weight corresponding to *v*, as shown in Equation (9),
(9)$$\;head_{i}=softmax\left(\frac{q_{i\;⊙\;}k_{i}^{T}}{\sqrt{d_{k}}}\right)v_{i},$$
where ⊙ denotes the dot product operation, $\sqrt{d_{k}}$ is used to scale the dot into the result to make the gradient of the model more stable, and *softmax* is used calculate the attention score by normalization, as shown in Equation (10).
(10)$$softmax\left(\cdot \right)=\frac{exp\left(\cdot \right)}{\sum exp\left(\cdot \right)}\;$$

Finally, the output of MHA is obtained by concatenating the attention results of the obtained individual head vectors, as shown in Equation (11).
(11)$${\mathrm{MHA}}_{out}=\mathrm{Concat}\left(head_{1},\cdots ,head_{h}\right)W_{O}$$

### 2.2. Discriminator

The design of the discriminator is shown in Figure 1. The M-layer convolution network acquires the image features, and then the feature values are normalized between −1 and 1 by the tanh activation function. If the value obtained is greater than 0, it is determined that the reconstructed separated image is true; otherwise, it is false, as shown in Equation (12),
(12)$$value=tanh\left(Conv_{M}\left(\hat{s}\right)\right),$$
where the parameters of the convolution network are set to be the same as those of the generator. The convolution kernel size is *k*, step size is *t*, and the tanh activation function is an odd function centered at 0, as shown in Equation (13),
(13)$$tanh\left(value\right)=\frac{e^{v}-e^{-value}}{e^{value}+e^{-value}},$$
which improves the convergence speed compared to the sigmoid activation function.

### 2.3. Loss Function

The generator and discriminator of the Transformer-guided GAN play off of each other in training; the generator optimizes the network parameters to reconstruct a reconstructed separated image that can fool the discriminator, and the discriminator improves the ability to judge true and false by adjusting the parameters. The objective function of the Transformer-guided GAN is shown in Equation (14).
(14)$$L_{\mathrm{GAN}}=E_{s}\left[\mathrm{logD}(s)\right]+E_{x}[\log(1\;-\;D(G(x))]$$
$L_{\mathrm{GAN}}$ guarantees that the reconstructed image given by the generator matches the distribution of the source image, while the SCBIS task requires the separation of the corresponding source image according to the given mix image x, so to obtain the separated image corresponding to the mix image, we add a reconstruction loss $L_{Rec}\left(G\right)$ to the generator, as shown in Equation (15),
(15)$$G^{*}=\arg\underset{G}{\min}\underset{D}{\max}\left[L_{\mathrm{GAN}}\left(G,D\right)+L_{Rec}\left(G\right)\right],$$
where $L_{Rec}\left(G\right)$ calculates the mean squared error between the separated image and the corresponding source image, as shown in Equation (16).
(16)$$L_{\mathrm{Rec}}{(G)=E}_{s,x}[(s-{G(x))}^{2}]$$

## 3. Datasets and Experiment Settings

### 3.1. The MNIST Dataset

The MNIST dataset [21] is used to evaluate the proposed Transformer-guided GAN. To meet the experimental requirements, we select 1000 handwritten digital images randomly from each of the categories 0 to 9 of the original dataset, then classify them into two categories according to the rules of 0 to 4 and 5 to 9 as the source image sets of the training set, and during the training process, each mix image is obtained by randomly selecting one image from each source image set and mixing them up. The same approach is used for the test set, and 200 images from each category of the original test set 0 to 9 are selected and divided into the two categories of 0 to 4 and 5 to 9. In total, 1000 test images are obtained by mixing the images in the two source image sets one-to-one.

### 3.2. Bags–Shoes Dataset

For verifying the performance of the proposed model on more realistic images, we use the shoes dataset [22] and the bags dataset [23] to form the bags–shoes mix dataset. In this dataset, 5000 images are used as the source image set for the training set, and the remaining 1000 images are used as the source image set for the test set. As with the MNIST dataset, the mix images used in the training are obtained by mixing images selected randomly from the two source image sets respectively in the training set. Similarly, the test set of mix images is obtained by one-to-one mixing of images from the two source image sets in the test set.

### 3.3. Experimental Settings

The Transformer-guided GAN model is implemented based on TensorFlow version 2.1. For the network parameters mentioned above, we have made the following settings. We resize mix images and source images to 64 × 64 × 3 and normalize them before the experiment. The generator uses a 4-layer U-shaped structure to connect the convolution and deconvolution networks with 16, 64, 256, and 768 channels per layer, respectively; the size of the convolution kernel is 4, the step size is 2, and the zero padding is required during convolution. In Transformer, the linear projection maps the convolution features into 16 patches of 768 dimensions using 3 Encoder modules and 4 head vectors. The Transformer-guided GAN was trained with a learning rate of 1 × 10^−4^ Adam optimizer, with 50 iterations. To ensure the fairness of the experiments, the same training parameter values were used for all experiments.

## 4. Results

We compare quantitatively and qualitatively the proposed Transformer-guided GAN with the frontier algorithms FastICA [2], NMF [4], Neural egg separation (NES) [24], AGAN [12], and PDualGAN [11] to indicate its superiority on SCBIS tasks. To ensure a fair comparison, the AGAN has the same UNet-GAN structure parameters as configured in this paper, except that the channel reduction factor k is set to 8 in the self-attentive module; the PDualGAN consists of 2 DualGANs, each using 2 UNet-GANs in the same configuration as this paper to implement mix to source mapping. In addition, to demonstrate the effectiveness of the combination of Transformer structure and UNet for the model, we designed ablation experiments.

### 4.1. Qualitative Results 

Figure 5 and Figure 6 show the qualitative results for the MNIST and bags–shoes datasets in the noiseless and noisy cases, respectively, from which it is clear that in any case, for both datasets, the quality of the separated images from our model is much higher than that of the other models. 

Qualitative results from the horizontal comparison of methods on the same dataset show that the FastICA algorithm barely separates the source images, which affects its metrics. The separation results given by the NMF algorithm show that the source image information is enhanced, but its loss in color, brightness, etc. leads to its unsatisfactory metrics. NES reduces the interference source information while leading to an increase in the disparity between the separated image and the source image in terms of structure and texture. The separated images of the AGAN from the MNIST dataset, where the source information is relatively simple, retain the information of the source images while eliminating the interference, although there are negligible artifacts. However, on the information-rich bags–shoes dataset, this interference cannot be neglected. The separation result of PDualGAN is similar to that of the AGAN but slightly inferior in the processing of more detailed information. In contrast, our proposed model is superior in terms of the overall structure, brightness, color, and detailed information processing.

From a longitudinal comparison of the qualitative results of the same method in different noise situations, we can see that the performance of all methods mentioned is significantly worse in noisy environments compared to noise-free results. The MNIST dataset, with its simple graphical structure, is less affected by noise than the bags–shoes dataset, with its rich detail. In addition to the traditional FastICA and NMF algorithms, methods using the GAN structure can filter the environmental noise and thus perform better in noisy situations. Furthermore, our proposed method gives better quality results than other methods in the noisy case, showing greater noise immunity.

### 4.2. Quantitative Results 

We used both PSNR and SSIM metrics to quantitatively compare the separation results. PSNR [25] calculates the peak signal-to-noise ratio between the separated image and the source image, and the higher the ratio, the higher the quality of the separated image, as shown in Equation (17),
(17)$$\mathrm{PSNR}=10\log_{10}\frac{{\mathrm{MAX}}_{I}^{2}}{\mathrm{MSE}},$$
where ${\mathrm{MAX}}_{I}$ is the image pixel maximum value, and MSE represents the mean variance between source image and separated image. SSIM [25] is used to evaluate the similarity between the source image and the separated image, as shown in Equation (18),
(18)$$\mathrm{SSIM}\left(s,\hat{s}\right)=\frac{\left(2\mu_{s}\mu_{\hat{s}}+c_{1}\right)\left(2\sigma_{s\hat{s}}+c_{2}\right)}{\left(\mu_{s}^{2}+\mu_{\hat{s}}^{2}+c_{1}\right)+\left(\sigma_{s}^{2}+\sigma_{\hat{s}}^{2}+c_{2}\right)},$$
where $\mu_{s},\mu_{\hat{s}}$ represent the mean values of s  and $\hat{s}$, respectively; $\sigma_{s}^{2}$, $\sigma_{\hat{s}}^{2}$ are the variance of s and $\hat{s}$, respectively; $\sigma_{s\hat{s}}\;$ is the covariance of s  and $\hat{s}$ ; and $c_{1}$, $c_{2}$ are constants designed to maintain stability, such that $c_{1}={\left(k_{1}L\right)}^{2}$, $c_{2}={\left(k_{2}L\right)}^{2}$, where L represents the dynamic range of pixel values, and $k_{1}$, $k_{2}$ are empirical values, where $k_{1}=0.01$, and $k_{2}=0.03$. SSIM takes a value between 0 and 1. The closer it is to 1, the more similar the separated image is to the source image.

Table 1 shows that the quantitative calculations are consistent with the results of the qualitative analysis, with our model achieving higher PSNR and SSIM metrics for the separated images than the other methods for the two datasets and their noisy cases. 

Horizontal comparisons of quantitative calculations show that our method outperforms the state-of-the-art AGAN by 7.7 dB PSNR and 0.07 SSIM on the simple MNIST dataset, while its PSNR in the noisy case is 6.3 dB higher than that of the AGAN, and its SSIM is 0.08 higher than that of the AGAN. On the bags–shoes dataset, with its more complex structure and details, the PSNR and SSIM of the Transformer-guided GAN are 7.7 dB and 0.06 higher, respectively, than those of the AGAN, and its PSNR and SSIM in noisy cases are 3.2 dB and 0.04 higher, respectively, than those of PDualGAN. The main reason is that, compared to the AGAN and PDualGAN, the Transformer-guided GAN can extract more detailed features through positional encoding and multi-headed attention.

A longitudinal comparison of the quantitative results shows that the PSNR and SSIM decreased in the noisy environment for all the methods mentioned. This is because the noise affects the extraction of source information by the models. The PSNR and SSIM of the AGAN in the noiseless case are 0.9 dB and 0.01 higher than those of the PDualGAN on the MNIST dataset, and similarly 0.9 dB and 0.01 higher on the bags–shoes dataset. This is due to the fact that the self-attention mechanism in the AGAN can capture detailed information. In the noisy environment, the AGAN and PDualGAN are close in qualitative results on the MNIST dataset, but on the bags–shoes dataset, PDualGAN is 1.3 dB and 0.03 higher than AGAN in PSNR and SSIM, respectively. This is because convolutional networks are more noise resistant than attention mechanisms for linear computing. Our model replaces self-attention with Transformer in the AGAN, and in combination with UNet, the PSNR and SSIM drop by 1.8 dB and 0.02 for the noisy MNIST dataset, but are still higher than the metrics of other methods in the noiseless environment, and drop by 7.8 dB and 0.07 for the noisy bags–shoes dataset, but were still comparable to the other metrics in the noiseless environment.

### 4.3. Ablation Test

We designed three comparison GANs for the ablation experiments: (1) a Transformer-based generator, (2) a generator using only UNet, and (3) a generator with the Transformer–UNet structure proposed in this paper, with the exception that the rest of the network structure and experimental setup remained unchanged. 

The quantitative results of the separation effects of the three networks on the two datasets are given in Table 2, respectively, showing that without UNet, the separation ability of the network decreases sharply and loses global control, while without the Transformer structure, although usable separation results can be obtained in mix images with a simple structure, there is the problem of not being able to ignore the details for information-rich source images; therefore, our model combines the two to improve the separation ability of the network in the face of complex structure images.

## 5. Conclusions

In this paper, a GAN structure based on the combination of UNet and Transformer is proposed to address the loss of detail information caused by ignoring location g correlation in SCBIS. UNet performs feature extraction on the source, Transformer acquires the detailed representations, and the overall reconstruction and detail reconstruction are achieved through GAN training. Experimental and comparative results show that the Transformer-guided GAN provides better quality separated images and more detailed representation of source information in the SCBIS task. For future research, there is ample scope for unsupervised blind image separation, as blind image separation in practice may not have a clean and usable source image on which to base the discriminator’s judgement. For future research, there is great scope for unsupervised SCBIS, as in practice, there may not be a usable source image.

## Figures and Tables

**Figure 1 sensors-23-04638-f001:**
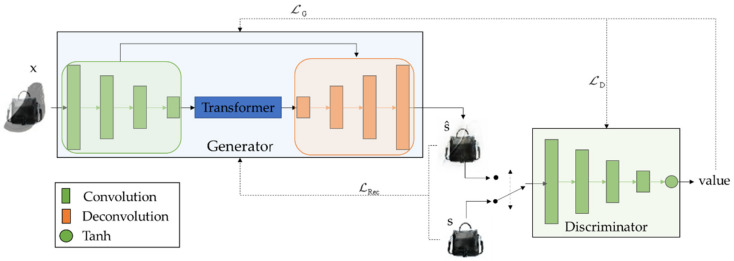
The overall structure of Transformer-guided GAN. The generator is used to reconstruct the separated image, and the discriminator to distinguish its authenticity.

**Figure 2 sensors-23-04638-f002:**
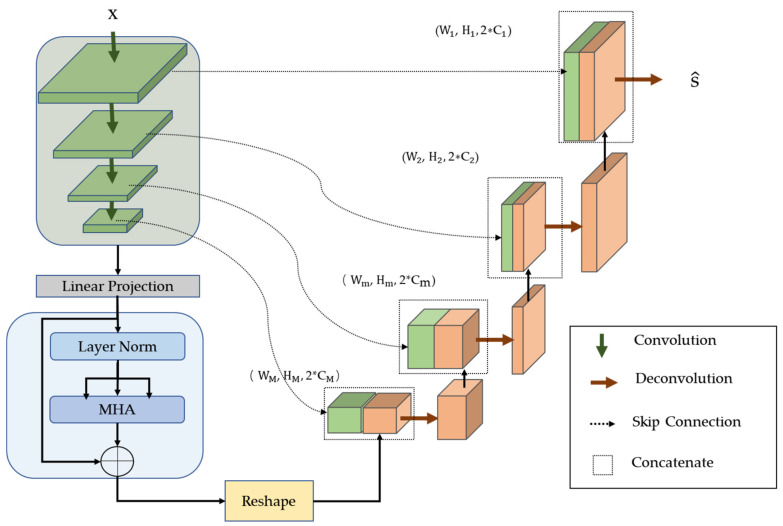
The detailed structure of the generator.

**Figure 3 sensors-23-04638-f003:**
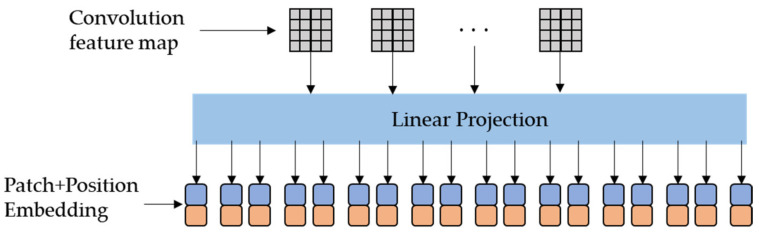
Mechanism of linear projection.

**Figure 4 sensors-23-04638-f004:**
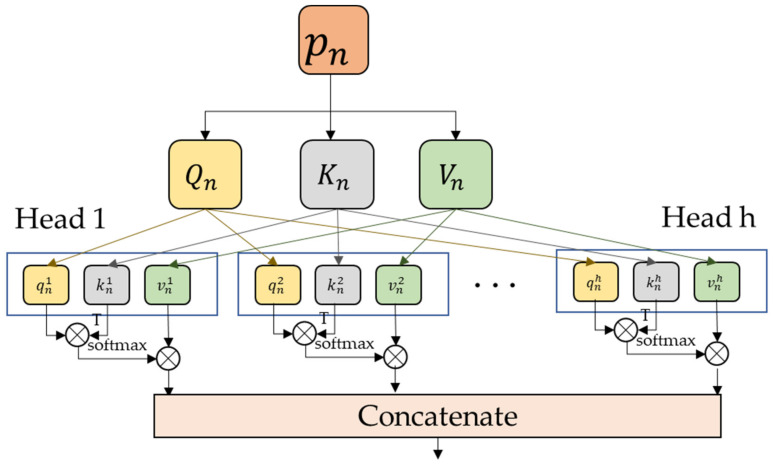
The structure of muti-head attention.

**Figure 5 sensors-23-04638-f005:**
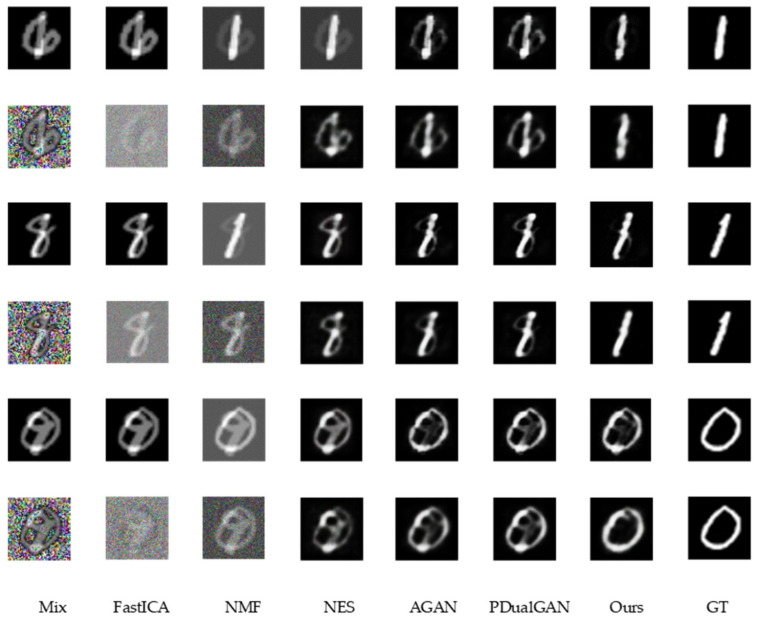
The qualitative results on MNIST dataset.

**Figure 6 sensors-23-04638-f006:**
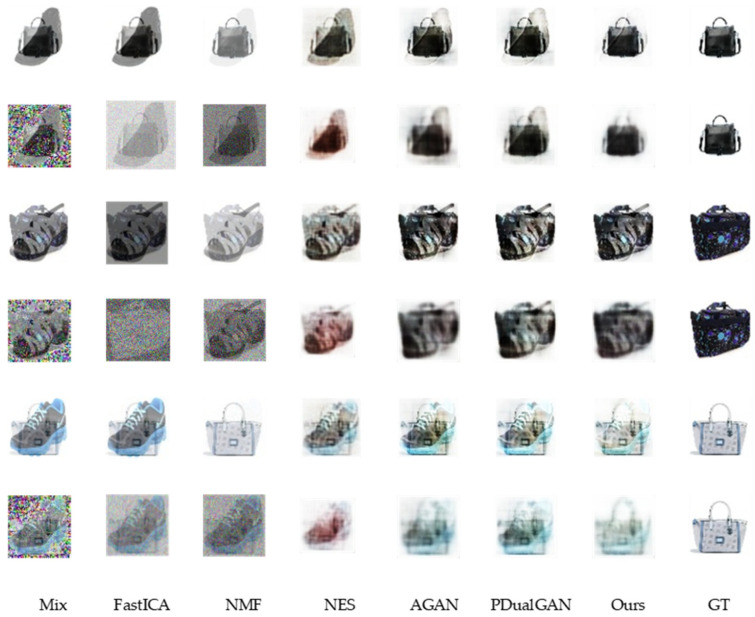
The qualitative results on bags–shoes dataset.

**Table 1 sensors-23-04638-t001:** The quantitative results.

PSNR (dB)/SSIM	FastICA	NMF	NES	AGAN	PDualGAN	Ours
MNIST	14.8/0.18	18.5/0.36	21.5/0.79	25.2/0.87	24.3/0.86	32.9/0.94
Noisy MNIST	14.1/0.04	17.3/0.30	20.0/0.74	23.8/0.84	23.7/0.85	30.1/0.92
Bags–shoes	17.2/0.29	16.4/0.32	17.4/0.76	23.6/0.89	22.7/0.88	31.3/0.95
Noisy bags–shoes	16.0/0.10	16.0/0.26	12.8/0.62	19.0/0.81	20.3/0.84	23.5/0.88

**Table 2 sensors-23-04638-t002:** The ablation results.

PSNR (dB)/SSIM	Transformer-GAN	UNet-GAN	Ours
Bags–shoes	8.6/0.41	22.9/0.88	31.3/0.95
MNIST	11.1/0.30	24.6/0.86	32.9/0.94

## Data Availability

Not applicable.

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
