# Peer review of "Single-Channel Blind Image Separation Based on Transformer-Guided GAN"

_sensors, 2023, doi:10.3390/s23104638_

Round 1

Reviewer 1 Report

Paper is well written. Authors should make it more clear in introduction their contribution compared to prior work

Author Response

Thank you for your  comments our manuscript entitled “Single-Channel Blind Image Separation Based on Transformer-Guided GAN”(ID:sensors-2293461). Those comments are all valuable and very helpful for revising and improving our paper, as well as the important guiding significance to our researches. We have studied comments carefully and have made correction which we hope to meet with approval. Please see the attachment.

Reviewer 2 Report

This article deals with an exciting topic on image processing, especially the blind image separation scheme. The author has done an extraordinary job in this study. It is appreciated. However, a few suggestions are given to enhance the quality of this manuscript.

Comments:

In the abstract, please avoid directly using acronyms, e.g.) UNet, consider expanding it for the first time.

Throughout the paper, a quick grammar and typo check is recommended.

In section 4, even though the results and tabulation are done, an intense discussion is required to support the claims. Please include some comparative analysis.

In section 4.1, Qualitative results, the paragraph starts with "The quantitative comparison results... given in Figure 5 and 6", but the figures are entitled as qualitative. Please avoid this ambiguity.

In conclusion, it looks so simple. Please include the achievements of your study and state the outcomes. Also, it is suggested to include the future scope that motivates the readers to carry out further.

Author Response

Thank you for your  comments our manuscript entitled “Single-Channel Blind Image Separation Based on Transformer-Guided GAN”(ID:sensors-2293461). Those comments are all valuable and very helpful for revising and improving our paper, as well as the important guiding significance to our researches. We have studied comments carefully and have made correction which we hope to meet with approval. Please see the attachment

Reviewer 3 Report

This paper proposes a Transformer-Guided GAN model guided by an attention mechanism for single-channel blind image separation. And the performance has been validated by quantitative experiments. Comments are listed below:

1. The literature review in Section 1 should be strengthened by providing more insights into the existing research and state-of-art archievement, with more attention to their limitations motivating the presented work.

2. How to determine the struture and training hyper-parameters in compared methods? Some necessary descriptions should be explained on compared models.

3. The exact calculation equations of PSNR and SSIM are recommended to be ilustrated directly.

4. How model performs under noisy environment?

5. Some minor errors. What's the exact meaning of "BIS", "SCBIS", "NES" and some other acronyms in this paper?

Author Response

Dear Reviewer:

Thank you for your  comments our manuscript entitled “Single-Channel Blind Image Separation Based on Transformer-Guided GAN”(ID:sensors-2293461). Those comments are all valuable and very helpful for revising and improving our paper, as well as the important guiding significance to our researches. We have studied comments carefully and have made correction which we hope to meet with approval. Please see the attachment

Round 2

Reviewer 3 Report

Much improvements have been done to improve the quality of the manuscript, However, some issues are still required to be addressed.

1. More state-of-art archievements should be discussed. At present, only 2 papers after 2021 are referred. As we know, much interesting works have been done based on Transformer model, as well as UNet-GAN one, recently.

2. How about the exact training configurations on compared AGAN and PDualGAN models? They should be present directly or with necessary reference statements in the manuscript. 

3. Some minor errors still exist. For example, double dot in Line 52 Page 2, Line 218 Section 4.1, Line 238 Section 4.1, Line 242 Section 4.1, . Moreover, what's the exact meaning of "excessive attention on its information" in "however, the attention model only focuses on the importance of single pixel value, leading to excessive attention on its information." at Section 1.1.1. "insufficient number of training samples" is recommended to revise as "insufficient training samples". The authors should revise the manuscript carefuly.

Author Response

Dear Reviewer:

Thank you again for your comments on our manuscript entitled “Single-Channel Blind Image Separation Based on Transformer-Guided GAN”(ID:sensors-2293461). Those comments are all valuable and very helpful for revising and improving our paper, as well as the important guiding significance to our researches. We have studied comments carefully and have made correction which we hope to meet with approval. We mark revisions and answers in red font. Please see the attachment for the main corrections in the paper and the responds to the reviewer’s comments 
